METHODS AND RESOURCES

# ReptiLearn: An automated home cage system for behavioral experiments in reptiles without human intervention

Tal Eisenberg[1], Mark Shein-Idelson [1,2]*

1 School of Neurobiology, Biochemistry, and Biophysics, The George S. Wise Faculty of Life Science, Tel Aviv University, Tel Aviv, Israel, 2 Sagol School of Neuroscience, Tel Aviv University, Tel Aviv, Israel

* sheinmark@tauex.tau.ac.il

## Abstract

Understanding behavior and its evolutionary underpinnings is crucial for unraveling the complexities of brain function. Traditional approaches strive to reduce behavioral complexity by designing short-term, highly constrained behavioral tasks with dichotomous choices in which animals respond to defined external perturbation. In contrast, natural behaviors evolve over multiple time scales during which actions are selected through bidirectional interactions with the environment and without human intervention. Recent technological advancements have opened up new possibilities for experimental designs that more closely mirror natural behaviors by replacing stringent experimental control with accurate multidimensional behavioral analysis. However, these approaches have been tailored to fit only a small number of species. This specificity limits the experimental opportunities offered by species diversity. Further, it hampers comparative analyses that are essential for extracting overarching behavioral principles and for examining behavior from an evolutionary perspective. To address this limitation, we developed ReptiLearn—a versatile, low-cost, Python-based solution, optimized for conducting automated long-term experiments in the home cage of reptiles, without human intervention. In addition, this system offers unique features such as precise temperature measurement and control, live prey reward dispensers, engagement with touch screens, and remote control through a user-friendly web interface. Finally, ReptiLearn incorporates low-latency closed-loop feedback allowing bidirectional interactions between animals and their environments. Thus, ReptiLearn provides a comprehensive solution for researchers studying behavior in ectotherms and beyond, bridging the gap between constrained laboratory settings and natural behavior in nonconventional model systems. We demonstrate the capabilities of ReptiLearn by automatically training the lizard *Pogona vitticeps* on a complex spatial learning task requiring association learning, displaced reward learning, and reversal learning.

## Introduction

Nervous systems evolved to facilitate behaviors enhancing survival and reproduction [1]. Understanding these behaviors and their evolutionary origins is crucial for unraveling the

**Data Availability Statement:** All relevant data are within the paper and its Supporting Information files. The ReptiLearn software and code are available from https://github.com/

EvolutionaryNeuralCodingLab/reptiLearn (DOI: 10.5281/zenodo.10546737).

**Funding:** This project has received funding from the Israel Science Foundation (ISF, grant No. 1133/20 to MSI) and the European Research Council (ERC) under the European Union's Horizon 2020 research and innovation programme (Grant agreement No. 949838 to MSI). The funders had no role in the study design, data collection and analysis, decision to publish, or preparation of the manuscript.

**Competing interests:** The authors have declared that no competing interests exist.

**Abbreviations:** IoU, Intersection over Union; SAM, Segment Anything Model.

complexities of nervous systems and the computational processes they support [2]. This endeavor was pushed forward by several experimental approaches. One approach is to use highly constrained behavioral tasks [3] that focus on a specific behavioral aspect (e.g., decision-making). Implementing such tasks requires reductionist experimental designs [4,5] that simplify behavioral complexities (e.g., a two-alternative forced choice task). This approach offers straightforward quantification, repeated trials that can be performed within laboratory settings, reduced variability within and between animals, and increased statistical power [3]. However, these benefits come at the expense of capturing only a limited subset of an animal's behavioral repertoire [6] and confining analysis to a specific task-dependent time scale. Further, this approach may introduce biases due to human handling and the use of tasks lacking the bidirectional interaction between animals and their environment as observed in the wild. In such bidirectional interactions, animals constantly receive information from the environment (e.g., perceive a prey) and act upon it (e.g., attack the prey), but at the same time, their actions change the environment (e.g., their prey escapes due to their approach [7]).

Recent technological advances offer a new opportunity to study natural behaviors in the lab or perform complex behavioral quantification in natural settings [2]. Miniaturization of cameras, automation of natural stimuli and reward delivery, and powerful computational tools now enable researchers to create autonomous environments that more closely resemble natural conditions [8–10] or home cage conditions [11–15]. In such settings, animals can be placed in complex environments [16,17] facilitating bidirectional interaction [17] and studied across multiple time scales [18,19]. New signal processing methods facilitate automatic annotation and analysis of the vast amounts of behavioral data collected in each experiment, resulting in a broad range of extracted behavioral features and improved statistical power [20]. These rich data sets offer a window into the variability inherent in animal behavior [6]. Importantly, many of these solutions are provided as open-source low-cost hardware and software packages, making them accessible to a wide range of research labs [21].

While the behavioral approaches above chart a promising path forward, they do not offer a comprehensive solution for many experimental scenarios. Primarily, they have been tailored to a limited number of species. Specifically, since the 1980s, a handful of genetically tractable model systems began to increasingly dominate scientific studies [22–24]. Correspondingly, automated home-cage monitoring and behavioral setups have been developed primarily for mice [15,19,25,26], fruit flies [27], and zebrafish [16], and offer species-specific behavioral paradigms [28]. This specificity extends to the devices used for interacting and rewarding the animals. For example, experimental systems usually lack temperature control and their automatic food dispensers are optimized for delivering dry food pellets rather than live insects. These properties limit applicability to nonconventional animal models such as reptiles and amphibians.

Reptiles and amphibians are large and diverse animal classes with many species offering unique perspectives on various biological and evolutionary research questions [23,29–32]. In recent years, research on these classes has gained momentum [33–35], aided by new genetic methodologies for probing [36] and manipulating gene expression [37–39]. In contrast to these advancements, implementation of new methodologies for studying behavior is limited. While progress has been achieved in automating specific elements within behavioral tasks for ectothermic vertebrates, or in adapting paradigms designed for other vertebrates [40–42], no complete solution exists for automated behavioral experiments in any reptile or amphibian. Correspondingly, the cognitive capacities of these animal classes remain poorly understood and research linking behavior with neurophysiology is scarce [43,44]. This deficiency can be attributed in part to the challenges posed by studying ectothermic vertebrates.

Ectothermic behavior is highly dependent on environmental temperatures and heat sources [45]. Thus, reducing unexplained behavioral variability necessitates continuous monitoring

and control of thermal conditions during experiments. Furthermore, ectothermy favors survival strategies with low energy consumption that manifests in increased immobility [46]. Such behaviors yield sparser behavioral data and require continuous long-term experiments with a sufficient sample size for robust statistical analysis. In contrast, most experimental approaches are not suited for continuous experiments over weeks and lack paradigms for bidirectional interactions with animals during such long periods. Another challenge for behavioral research in ectothermic vertebrates is reinforcement using food rewards. These animals can often go without food or drink for extended periods compared to mammals [47,48], potentially reducing their motivation and engagement in behavioral tasks. Moreover, most automated systems employed in mammalian research make use of liquid rewards [13,17,19], which are unsuitable for species requiring live prey as a reward. Conversely, reinforcers such as heat can be utilized but are not included in automated behavioral systems. Thus, human intervention is required during experiments, making it more difficult to increase sample sizes and reduce variability.

To address these issues, we developed ReptiLearn—a new comprehensive solution for behavioral experiments in reptiles (Fig 1A). This platform includes unique hardware components such as a fine-grained temperature control and measurement apparatus, automated live feed reward dispensers, a touch screen for providing visual stimulation and logging touch choices, and modules for interacting with Arduino components (Fig 1B). ReptiLearn is ideally suited for continuous experiments over extended time scales with the arena serving as the animal's home cage for days to weeks. Arena components are controlled by a dedicated software suite allowing flexible design of fully automated experiments and extraction of behavioral features (Fig 1B). These experiments can be controlled remotely by a web-based user interface for increased accessibility (Fig 1D). Finally, to facilitate automated bidirectional interactions of animals with their environment, ReptiLearn incorporates low-latency components that allow closing a loop between behavioral dynamics and arena hardware. We demonstrate ReptiLearn's capabilities by successfully training *P. vitticeps* lizards on an automated spatial learning task.

## Results

### Real-time movement tracking

Locomotion and posture changes provide valuable information about the animal's behavioral states and strategies [49] and are linked to environmental factors such as temperature [50]. Further, animal movements are critical components of the bidirectional interactions between animals and their environment [7]. To continuously log movements and link them in real time to arena apparatus (e.g., delivery of food or visual stimuli), we implemented in ReptiLearn 2 video-based movement tracking algorithms (Fig 1). In the first, our aim was to optimize processing speed in real-time experiments. To do so, we trained a light-weight neural network (YOLOv4, [51]) that can detect the position of the animal's head bounding box (Fig 2A). YOLOv4 offered a good tradeoff between accuracy and computation time. The latency to position detection was narrowly distributed (Fig 2B) with a mean of 7.92 ms (SD = 0.36 ms), which is low enough for real-time tracking of every frame in streams of up to 125 Hz. We also achieved good accuracy (Figs 2C and S1) with a position detection error of 0.73 cm (SD = 0.91), a recall of 100% (no false positives), and a precision (true positive/total positive) of 78% (Fig 2D). To estimate the latency in closed-loop experiments, we measured the time between turning on an LED in the arena and its detection in the video stream. This delay added 43 ms (SD = 9.34 ms) to processing time (Fig 2E), resulting in a total delay of 50.92 ms (SD = 9.35 ms) for location based closed-loop feedback.

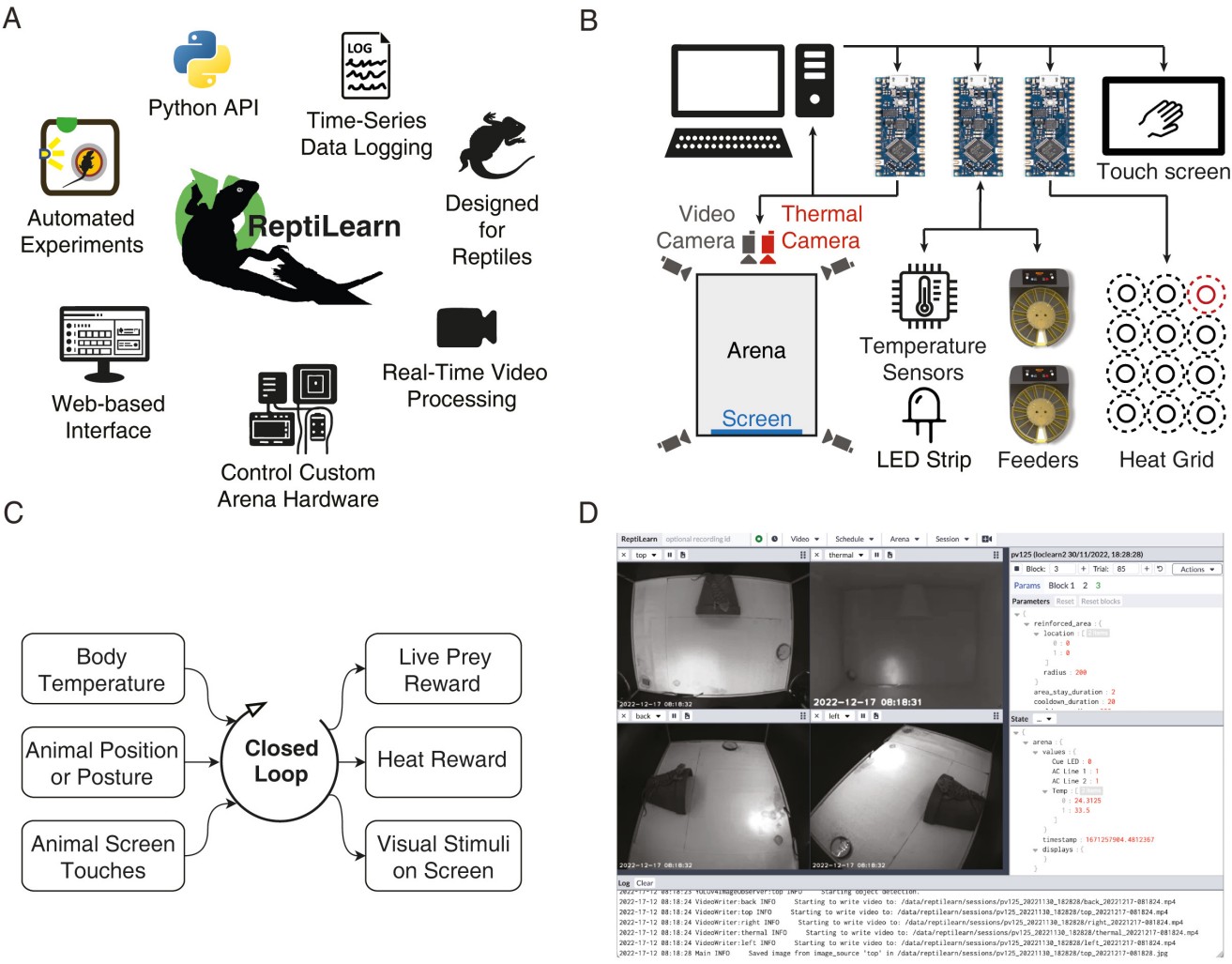

**Fig 1. Features and design of the ReptiLearn behavioral system.** (**A**) A schematic of the features supported by ReptiLearn. ReptiLearn is written in Python, provides an API for automating experimental tasks, runs real-time processing, controls arena hardware components including live food dispensers and a heat reward system, collects time-series data, features a web-based user interface for remote monitoring and control. (**B**) Diagram of hardware components in ReptiLearn. The arena includes synchronized visual and thermal cameras, temperature sensors, live prey feeders, a grid of 12 heat lamps covering the arena, illumination LEDs and a touchscreen. The hardware is controlled using Arduino boards and designed with generic interfaces for diverse research needs. ReptiLearn can run with different subsets of the above components. (**C**) A schematic illustrating the real-time closed loop processing in ReptiLearn. ReptiLearn allows implementing closed-loop behavioral tasks linking any of the following features in real-time: Animal and ambient temperature, animal position and posture, animal screen touches, live prey or heat reward, and visual stimulation on the screen. (**D**) Screenshot of the web-based user interface. The interface allows monitoring the cameras (top left) and the status of hardware (state panel on the right) as well as controlling the arena (top menu) and experiments remotely (experimental design panel on the right). Experiment events and system information appear in the log (bottom).

While the above approach provides a fast solution for real-time feedback, it could be easily replaced with other models if computational time constraints are loosened [20,52]. To acquire richer position and posture information, we used a second approach in which we fed the head bounding box calculated by YOLOv4 to an object segmentation model (Segment Anything Model, SAM; [53]). This model provides a mask of the entire animal body (Fig 3C). The processing time of this model is much longer, making it less practical for low-latency real-time applications. Its advantage, however, is that it does not require additional training or manual annotation and can provide rich data about the animal's behavior (as later shown).

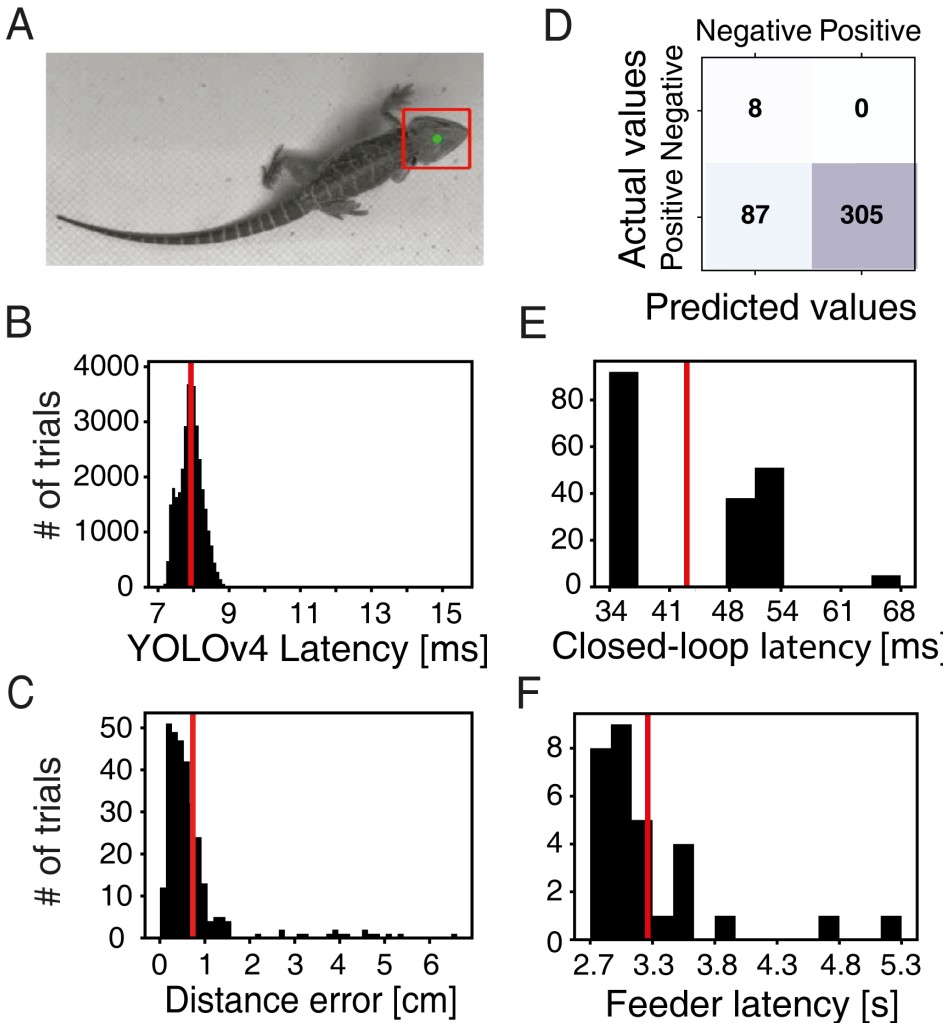

**Fig 2. ReptiLearn exhibits high performance with low latencies. (A)** Example of the fine-tuned YOLOv4 model output. The model was trained to detect the bounding box of the lizard's head (red). The box centroid (green) was used to estimate the animal's position. **(B)** YOLOv4 object detection latency distribution. Latencies were measured between the time of receiving the image by the model and the output time of position calculation. **(C)** Distribution of Euclidean distances between YOLOv4 bounding box center and annotated ground truth over a validation set of 400 images sampled uniformly from video data of 4 animals. Notice small distance errors (mean = 0.73 cm, SD = 0.91). **(D)** A confusion matrix showing the results of the fine-tuned YOLOv4 model over the validation set. Model was tuned to have a zero false positive rate. **(E)** Distribution of latencies in closed-loop experiments. Latencies were measured from the time of sending a command to turn on an LED to the time of detecting LED intensity change in the video stream (this delay comprises the arena controller, video acquisition, and LED detection analysis). **(F)** Food dispenser latency distribution. Latencies were measured from the time of sending a command to the worm dispenser to the detection of the dispensed worm in the video stream after landing on the arena floor. Rewards are received within an average time of 3.21 s (SD = 0.54 s). Individual numerical values are provided in S1 Data.

## Devices for closed-loop long-term automated interactions with animals

Automating long-term experiments require integrating components for animal interactions and providing the necessary conditions for survival. The controlled delivery of these conditions can also serve as reinforcement in behavioral tasks. For that aim, we integrated modules for closing the loop between behavior (movement) and arena apparatus such as visual stimuli, heat delivery, and live food reward. The latter 2 being most relevant for reinforcing behavior in reptiles and amphibians [54,55].

To integrate automatic delivery of live food (e.g., *Tenebrio molitor* larvae), we modified inexpensive aquarium fish feeders (Methods) to achieve a narrow distribution of reward delays with mean latency of 3.21 s (SD = 0.54 s) (Fig 2F). If shorter delay times are needed, visual stimulation devices (single LED illumination, arena LED strip illumination, or stimulation on a touch screen) were implemented for bridging the gap between movement detection and reward delivery. For presenting images or videos, we integrated into ReptiLearn a web application that can be used to display and animate custom stimuli on any number of touch screens (Fig 1B). Animal screen touches can be registered and relayed back to the system for real-time feedback to the displayed stimuli or to other triggerable arena components (Fig 1C). This module can be used, for example, for displaying prey items on the screen and for logging lizard screen strikes.

The ability to spatiotemporally control arena temperatures is also important for survival and for reinforcing behavior in ectotherms [54]. To achieve such control, we integrated a grid of infrared heat lamps (Fig 1B) that could be independently and automatically turned on (Methods). The heat lamp coverage provides fine-grained control over the arena's thermal gradient (Fig 3A), thus allowing to test temperature preference and thermal regulation under flexible spatial configurations. In addition, these heat lamps, in contrast to ceramic heating elements, induce quick temperature changes with a detectable increase of 1°C on the arena

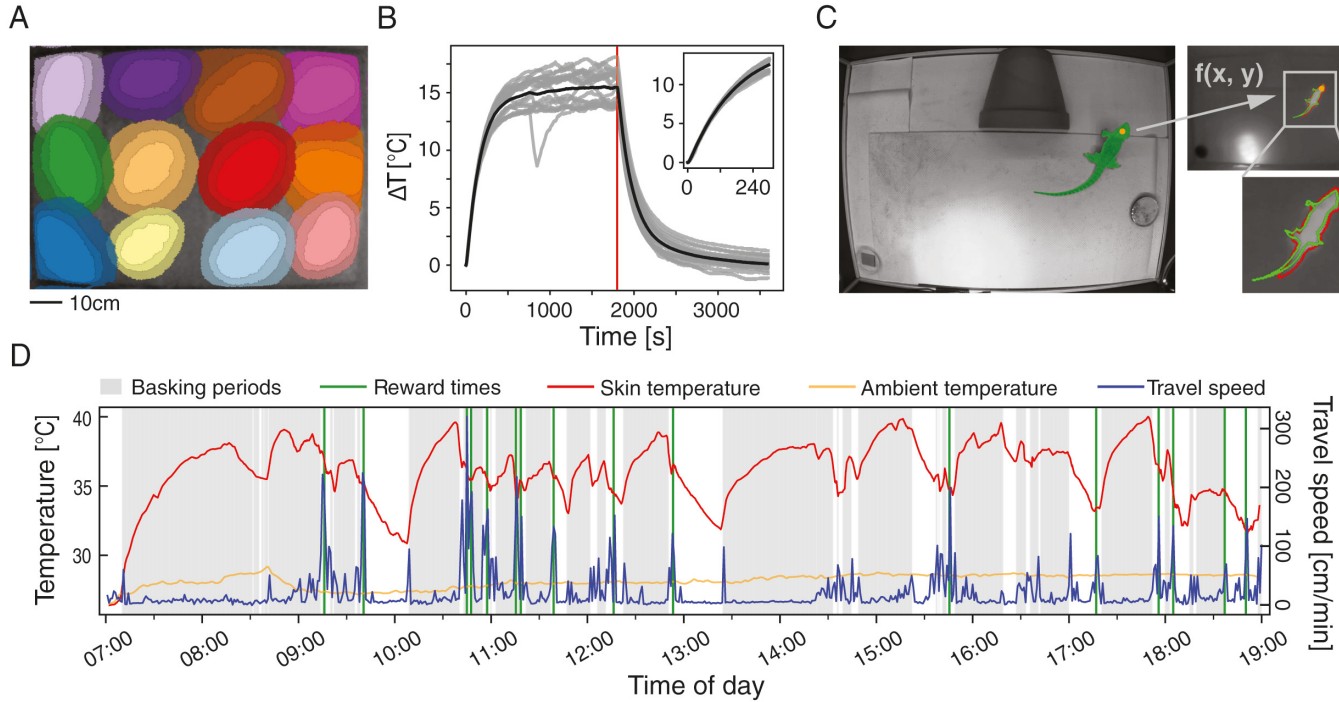

**Fig 3. ReptiLearn allows extracting complex relations between skin temperature and spatial dynamics. (A)** Heat gradients of a grid of 12 heat lamps across the arena floor. The outer, middle, and inner areas of each lamp represent temperature increases of 4°C, 6°C, and 8°C above baseline, respectively. **(B)** Temperature dynamics of a single heat lamp measured with a digital temperature sensor over trials (gray) and on average (black) relative to initial temperature (0). The lamp was turned off after 30 min (red line). **(C)** Skin temperatures were calculated by first segmenting the animal in the regular camera image (top, green) using the SAM algorithm, based on the output centroid of the YOLOv4 model (orange dot). Next, a linear function *f* is used to transform the mask and centroid to the thermal image pixel space (right top) and the skin temperature is determined by calculating the median over the mask (right bottom). A comparison with a SAM mask calculated directly from the thermal camera (red) is used to assess the validity of the transformed mask. **(D)** Movement dynamics (travel speed, blue) and corresponding skin temperature changes (red, with ambient temperature in orange) as well as reward times (green) and basking periods (gray) measured over a single day. Individual numerical values are provided in S1 Data.

floor within 18.11 s (SD = 1.7 s) of turning on the heat source (Fig 3B), and can thus provide an instantaneous reward without the need for manual refilling, as in the case of live insect rewards [56]. While temperature change sensitivity in *P. vitticeps* is unknown, reptiles are equipped with molecular machinery for sensing temperature changes smaller than 1˚C [57], presumably allowing much faster heat reward detection. Taken together, ReptiLearn integrates multiple methodologies for interacting with animals and for delivering rewards with a low-latency feedback, which are instrumental for effective learning and conditioning paradigms [54,58–60].

### Measuring the interplay between behavior and skin temperature

Integrating temperature control with body temperature measurements can open up many possibilities for studying temperature-dependent behavioral features such as thermal regulation [54,61]. Doing so requires continuous measurement of body temperature during long experiments. While measuring core body temperature requires internal probes, skin temperature is easily measurable using a thermal camera [62]. To measure skin temperature using an infrared camera, we first generated a segmentation mask of the lizard's body position using SAM (Methods) on the regular video camera stream (as described above; Fig 3C). We then transformed arena coordinates from the visual camera to the thermal camera and estimated median skin temperature across the mask (Methods). We validated the accuracy of the transformed body mask by showing a consistent temperature drop between the lizard's body and it is surrounding over thermal video frames (Methods; S2A and S2B Fig).

Combining measurements of animal temperature together with animal dynamics can shed light on thermo-regulation strategies. Fig 3D (red curve) shows temperature dynamics during a day of measurement together with travel speed (blue curve), calculated by integrating position changes over time. Periods of basking were detected (Fig 3D, gray shade) when lizards entered a circle of approximately 40 cm diameter under the heat lamp. This analysis shows that the lizard spent a large fraction of its time stationary. During most of this time, the lizard was under the heat lamp and increased its temperature. Occasionally, it was stationary in cold spots and decreased its temperature (Fig 3D, white areas). Between these stationary periods, the lizard exhibited short bouts of activity (Fig 3D, blue traces). This dynamic allowed the lizard to maintain an average preferred temperature of 36.0˚C (SD = 2.3˚C) that is considerably different from the ambient temperature (28.2, SD = 0.5˚C, orange curve), in general accordance with previous studies [61,63]. These data reflect skin temperatures but can be converted to core temperature using a simple calibration as previously performed for *P. Vitticeps* [64] (S2C Fig) and additional lizards of similar size [63,65,66]. Interestingly, while in some instances basking was observed following food reward (Fig 3D, green lines), this was not the general case hinting that the decision to bask after feeding is not always prioritized.

### An automated paradigm for spatial learning in *Pogona Vitticeps*

To demonstrate the capabilities of the system we trained lizards on a spatial task requiring association learning, displaced reward learning and reversal learning. Lizards were placed in a ReptiLearn arena, which served as their home cage for the duration of the experiment (lasting 2 to 3 weeks). The experiment consisted of 3 blocks. In the first block, lizards were required to enter the feeder dish area in order to receive live food reward from the automatic feeder in the rewarded location (Fig 4A). This block served to habituate the animal to the arena and to associate entering the dish location with a light blink and a food reward. In addition, to prevent frequent rewards without new action, animals had to exit the reward area (gray area outside the reward circle in Fig 4A) before entering it again to receive another reward. In the second

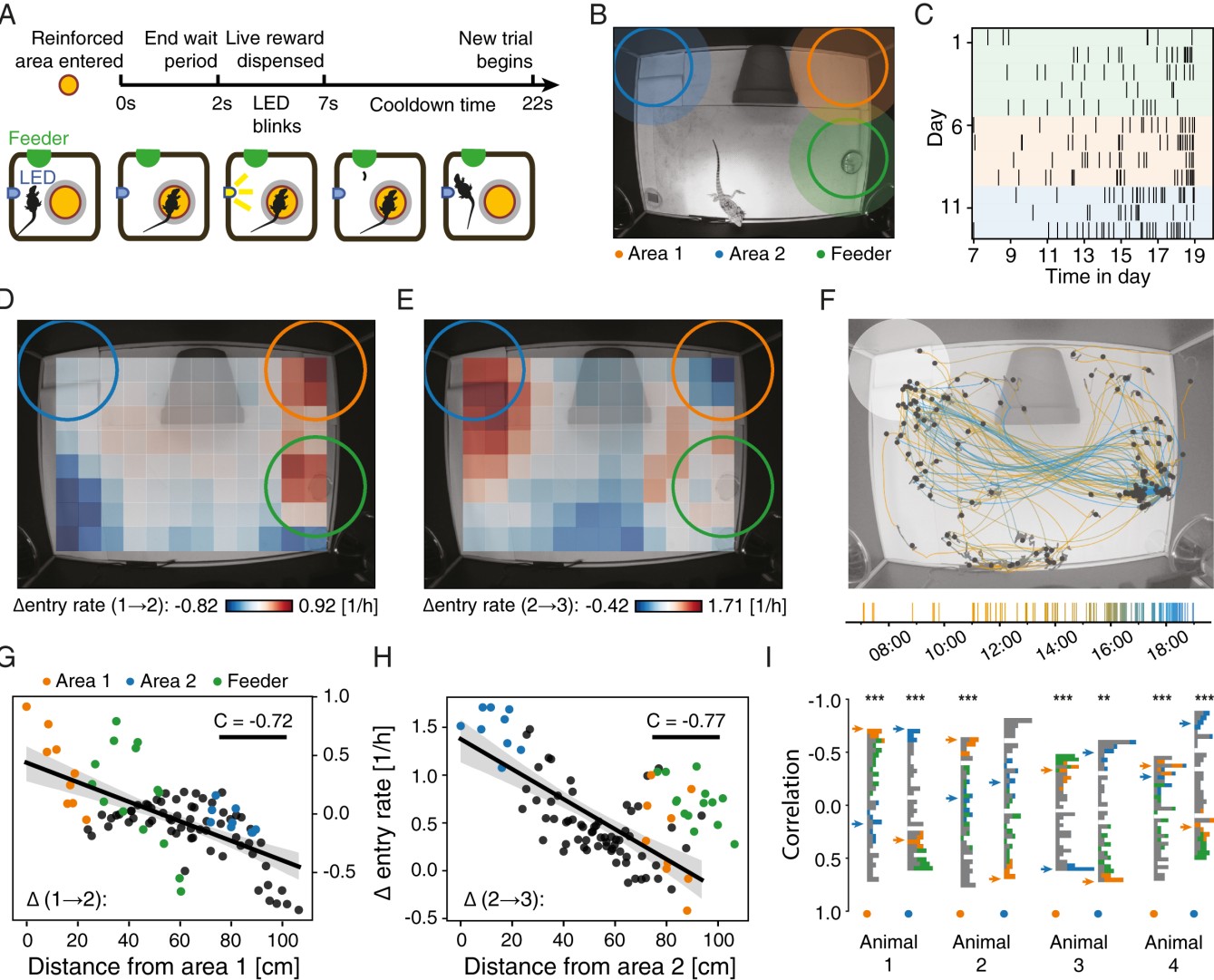

**Fig 4. Lizards learn a spatial task in a closed-loop automated paradigm.** (A) Schematics of a spatial learning task block. The lizard was conditioned to visit a predefined area in the arena (yellow circle, not physically marked in the arena and invisible to the lizard). Following the lizard's detection in the rewarded location (entering the rewarded location and staying there for 2 s), an LED was turned on (blinks for 5 s), after which a live prey was delivered. After each reward, the animal had to exit the cooldown area (gray circle), and 20 s had to pass before the lizard could return to trigger another reward. (B) The experiment consisted of 3 blocks (as in (A)). In the first, the feeder area (green) was reinforced with food reward. In the second and third blocks, the top-right corner (orange) and top-left corner (blue) were reinforced, respectively. (C) Raster plot of reward times for animal 1 over the full 12-day experiment. Background colors correspond to the reinforced areas. (D) The change (between the first and the second blocks) in the lizard's entry rate (ΔER) to different areas plotted on the physical space of the arena. Each square is a mean over entries to a circular area (same size as the reinforced area) surrounding the square. Notice the strongest increase for the second reinforced area but also a significant increase in the reward location. (E) Same as (D) but for the changes between the second and the third blocks of the same animal. Notice the strong increase for the third area (blue) and strong decrease for the second area. (F) Position trajectories over a single day (animal 1, block 3, day 3), segmented into stationary (gray dots marking mean location during stationary periods) and movement periods (colored lines). (G) ΔER (as in (E)) for each square as a function of its distance from the second reinforced area. Linear regression analysis shows significant ($p < 0.001$) correlation of C = −0.72. Areas overlapping with the feeder area (green points) were excluded from the regression. Blue and orange dots correspond to blue and orange areas in (B). (H) Same as (G) but for the increase between the second and the third blocks as a function of distance from the third reinforced area ($p < 0.001$; C = −0.77). (I) Distributions of correlation coefficients for ΔER as a function of distance (as in (G) and (H)) across areas for different animals and blocks (orange and blue dots correspond to blocks 1→2 and blocks 2→3, respectively). Areas in the histogram are color coded as in (G). Orange and blue arrows indicate the center of the reinforced areas of the second and the third blocks, respectively. Notice strong shifts in ΔER correlation distribution following the change in rewarded area for all animals, except the third block in animal 2. Individual numerical values are provided in S1 Data.

block, we repeated the same reward protocol but changed the reinforced area to the top right corner (Fig 4B, orange circle) without changing the location of the reward. We note that rewarded locations were only defined in the video analysis algorithm and were not physically marked in the arena. This block served to examine whether lizards can associate their action (entering a specific location), with an outcome (reward) in another location (displaced reward). In the third block, we examined if lizards could reverse their learning by changing the reward location to the upper left corner (Fig 4B, blue circle).

Lizards successfully entered the rewarded areas in each block and received rewards, all without any guidance or interaction with humans throughout the entire experiment. The reward rate was not homogenous over time (Fig 4C). Reward rate was low at the beginning of the day and increased during the afternoon. However, these dynamics were lizard specific with different lizards showing increased reward rates during different times (S3 Fig). To assess whether animals learned the task, or entered the rewarded areas by chance, we posited that significant change in the entry rate to a reinforced area implies an association between the area and the reward. We therefore tracked the entry rate to the reinforced area and to all other areas (uniformly distributed on a grid with centers marked by rectangles in Fig 4D) and calculated the differences in entry rate (ΔER) between consecutive blocks (Fig 4D and 4E). The first transition in reward location (block 1→2) was accompanied by an increased ΔER to the reinforced area (Fig 4D, orange circle). An increase in ΔER was also observed for the reward location (feeder area). This increase is expected since the lizard continues to receive the mealworms in the feeder location. To quantify the changes in the lizard's behavior, we plotted ΔER for each area as a function of the distance from the reinforced area and conducted a regression analysis (Fig 4G). In this analysis, we exclude the area around the feeder since the animals had to enter this area to receive rewards (Fig 4G, green dots). We expected an increase in ΔER to areas closer to the reinforced area. Such an increase was evident (Fig 4G, $p$-value < 0.001). Repeating this experiment in additional animals showed a similar and significant decrease in ΔER with distance (Fig 4I). This decrease evolved gradually over the training period but fluctuated during single days (S4A Fig). These results indicate that *P. Vitticeps* can learn to associate one location in the arena with an outcome in a different location.

We next examined if the lizards could perform a reversal learning task. After the lizards learned the first location, we shifted the reward location. Following the switch, we observed an increased ΔER in the new reinforced location (Fig 4E, blue circle). Furthermore, we observed a decrease in the previous reward location (Fig 4E, orange circle). This result corresponded with a significant ($p$-value < 0.001) spatial decay of ΔER with distance from the new reinforced location (Fig 4H) that developed gradually over days (S4B Fig). Correspondingly, examining all movement trajectories on the last day of training (Methods) revealed that lizards were engaged in stereotypical movement paths between the feeder and the rewarded location (Fig 4F). Successful learning in the reversal task was significant across animals (Fig 4I) with one animal failing to learn the reversal (see S1 Table for U values and statistical significance for all animals and blocks). However, this animal showed a strong decrease in movement in the third block which may explain its nonsignificant learning. Thus, using ReptiLearn we were able to successfully train lizards on a complex spatial task without human intervention.

## Discussion

In this manuscript, we introduce ReptiLearn—a versatile, low-cost, open-source experimental arena for behavioral experiments in reptiles (Fig 1A). As far as we are aware, this is the first comprehensive automated solution for behavioral investigations in reptiles. It effectively addresses numerous challenges inherent to behavioral studies in reptiles, amphibians, and

beyond. Specifically, ReptiLearn facilitates precise control and monitoring of arena and animal temperature. It incorporates a specialized feeder for delivery of live prey over long periods. In addition, the system is fully automated and offers a wide range of classes for controlling hardware components, which can be remotely controlled and monitored through a user-friendly web interface. Finally, ReptiLearn operates in real-time and permits the flexible coupling of arena components to design diverse experiments (Fig 1C). The ReptiLearn code, and a user-friendly installation procedure complemented by tutorials, is accessible at https://github.com/EvolutionaryNeuralCodingLab/reptiLearn.

While primarily designed with reptiles in mind, ReptiLearn offers innovative solutions that can be applied to experiments involving other animal models. The low-cost feeder, along with its control circuit and software, can prove valuable for conditioning species reliant on live prey, such as shrews [67] and insectivorous birds [68]. The web-based user interface (Fig 1D) is ideal for continuous, long-term experiments as it simplifies remote monitoring of multiple camera feeds and can be accessed easily via mobile devices like smartphones and tablets. Additionally, the highly parallelized image processing pipeline can efficiently scale with the number of CPU cores and GPUs used, allowing for tracking animals across multiple cameras, forming the basis for 3D tracking solutions. This can be particularly useful for combining image analysis data from different devices, for example, for estimating real-time skin (or body, S2C Fig) temperature and employing it as an input for closed-loop feedback (Fig 3). Combining SAM with object detection models, such as YOLO, for generating animal body segmentation masks presents a promising new approach that can save many hours of manual annotation and be used for analyzing animal behavior dynamics.

The fine-grained control and monitoring of both the arena's and animals' temperatures removes barriers when studying thermoregulation in ectotherms [62] and endotherms [69,70]. By utilizing advanced tools for identifying animals in video streams and automatically registering them to images captured by thermal cameras, we were able to track animal skin temperature continuously (Fig 3). This approach offers distinct advantages over traditional methods. Such solutions require surgical procedures for implanting temperature probes [64,66] and telemeters [61], the telemetry location information has a lower spatial resolution relative to video, and there is no access to posture information. Further, the monotonic relation between skin and core temperature (with approximately linear relation at temperatures of up to approximately 35˚C) allows estimating core temperatures from skin temperatures [63–66] (S2C Fig). Using such estimation, our approach could be used for measuring the preferred temperature of freely behaving animals without the need to construct specialized setups [71] by simply averaging over the animal's temperatures (Fig 3D). Further, our approach enables placing thermoregulation within a wider behavioral and neurophysiological context [2]: First, skin temperature information can be measured for any task the animal performs and can be combined with shuttling boxes when accurate linear gradients are required [71]. Second, tracking and manipulating food rewards allows incorporating metabolic considerations into experiments. Third, the arena is compatible with neurophysiological measurements allowing to link thermoregulation with brain activity [72]. Finally, the ability to dynamically alter thermal conditions using real-time feedback opens new avenues of thermal regulation research and significantly enhances the system's flexibility.

ReptiLearn is well suited for investigating short-range spatial cognition [73]. We demonstrated its efficacy in a complex spatial task involving *P. vitticeps* and encompassing association learning, displaced reward learning and reversal learning. In addition, we demonstrate that reptiles can use free-exploration without any guidance or human feedback to learn this task and flexibly associate specific unmarked positions with a reward in another location. Our results align with prior studies describing reptile spatial learning abilities [74–78]. Notably, our

innovative approach departs from previous spatial assays that necessitated performing an extensive number of trials, each lasting up to tens of minutes [58,77], until performance criteria were met [79]. These experiments were usually conducted manually, demanding substantial efforts from experimenters, such as baiting food rewards in each trial [76] or repositioning of animals [77]. In contrast, our paradigm allows unrestricted animal movement throughout the entire experiment eliminating possible biases introduced by human handling. Further, animals choose when to engage in the task as part of their uninterrupted behavioral routines. The lack of defined trials prevents calculating trial statistics but provides information about animal activity preferences (Fig 4F). This approach results in high engagement over consecutive days during which animals successfully achieved the task tens of times per day (S3 Fig). While this rate is hard to compare to paradigms with defined trials, the number of reward events is comparable and in many cases higher than in conventional trial-based experiments [75,79,80]. Finally, our automated long-term recording approach is ideally suited for ectotherms with low metabolic rates and behaviors that likely extend over long time scales. Another added benefit of this approach is collecting large statistical datasets. By harnessing continuous position tracking (Fig 4F), temperature manipulation and monitoring (Fig 3), visual stimulation (Fig 1B), and real-time feedback (Figs 1C and 2), ReptiLearn allows expanding the range of questions studied in reptile spatial cognition.

Despite pioneering work in the field [44,58,77,81,82], our understanding of the cognitive abilities of reptiles, and ectotherms in general, remains limited. Progress has been constrained, in part, by the lack of modern behavioral and neurophysiological tools suited for ectotherms. In this study, we demonstrated how ReptiLearn enables systematic investigation of spatial cognition and short-range navigation. This approach can be easily extended to explore additional cognitive capacities. For example, by presenting different auditory or visual stimuli on the screen and differentially linking them to rewards, we can gain insight into the sensory processing capacities of reptiles [83]. Temperature manipulation and monitoring during these tasks will allow linking cognitive performance with thermal regulation. Long-term performance monitoring can provide insight into memory prioritization and long-term storage capacities.

We demonstrated the usefulness of ReptiLearn in one species out of the large and diverse class of reptiles. While our approach will not fit all reptiles, many of the challenges we tackled are not unique to *P. Vitticeps*. The strong dependence on temperature, weak dependence on food and water reward, preference for eating live prey, low movement rates and engagement in repeated trials, and slow habituation to human handling are shared across many ectothermic vertebrates [43,58]. Thus, studies in a wide range of species can potentially benefit from our approach. Finally, ReptiLearn facilitates the integration of behavioral assays with neurophysiology to elucidate the brain regions and neuronal population patterns underpinning these cognitive skills [72]. Recent neuronal recordings from awake reptiles are beginning to reveal the neuronal dynamics underlying brain states [72,84,85] and visual processing [86]. Placing such studies within a behavioral context can shed light on the evolution of cognition and provide a comparative perspective critical for generalized understanding [29,30,32–35,81]. ReptiLearn substantially enhances the toolbox available to behavioral researchers studying ectotherms and serves as a proof of concept for an automated, minimally intrusive approach for exploring reptile cognition.

## Methods

### Animals

Four *P. vitticeps* lizards participated in the experiment. Two adults—a male (animal 1, 186g) and a female (animal 2, 231g), and 2 juveniles—male (animal 3, 121g), female (animal 4,

117g). Lizards were purchased from local dealers and housed in an animal house at Tel Aviv University's zoological gardens. Lizards were kept in a 12–12 h light (07:00–19:00) and dark cycle and a room temperature of 24˚C. All experiments were approved by Tel Aviv University ethical committee (Approval number: 04-21-055).

## Spatial learning task

**General task conditions.**   Animals were moved from the animal house to the arena near the end of the light period and remained there for the duration of the experiment (2 to 3 weeks). The arena contained a shelter, a water dish, and a basking area. Before starting the experiment, each animal was cleaned inside an open box filled with shallow water to prevent excessive dirt from reducing position-tracking accuracy. Subsequently, animals were placed in the dark arena, simulating nighttime, and the experiment protocol began the following day (at 7 AM). During all experiments, lizards received a vegetable meal every 7 days. They were given live food manually in case they did not reach a minimum of 10 worms every 3 days. The experiment was fully automated by the experiment python module (see system/experiments/loclearn2.py in the ReptiLearn repository). The LED strip and the heat lamp were automatically turned on from 7 AM to 7 PM daily.

**Task structure.**   The task consisted of 3 blocks. In the first block, lizards were rewarded for entering the feeder area, and in the second and third blocks for entering area 1 and 2, respectively ([Fig 4B]). Specifically, upon entering the rewarded location and staying there for 2 s (wait period), the cue LED started to blink and 5 s later it was turned off, and a command was sent to dispense the reward. In pilot tests, we found that lizards positioned themselves on the edge of rewarded areas such that slight movements resulted in repeated crossing into the rewarded areas. To solve this, we required animals to exit the cooldown area (and wait 20 s starting from LED blink onset) before entering the rewarded area again to receive another reward. Rewarded locations were only defined in the video analysis algorithm and were not physically marked in the arena. The transition between blocks occurred after 60 rewards were received, or after 6 days, whichever occurred first.

**Logging task data.**   Lizard position was constantly tracked throughout the task. Entrance to the rewarded location was determined in real time by the experiment module (see software) that received input from the YOLOv4 ImageObserver (see software) tracking the animal's head position from the top camera video stream. The ImageObserver output was stored in CSV files for later analysis (see software). The event logger tracked the time of reward releases, their resupply times, and the moments when the animal entered or exited the reinforced area. Switching between the 3 reinforced areas was done manually using the Web UI by moving to the next block in the session UI section (see software). Switching was always done at the beginning of a day so that the same area was rewarded throughout each day. The experiment module tracked the number of available rewards in each stacked feeder and switched between feeders automatically when one of them was empty. This information was also relayed to the experimenter through the Web UI so that the feeders could be manually refilled when necessary.

## Analysis

**Closed-loop latency test.**   We estimated the latency of delivering a closed-loop stimulus by combining the processing duration of the YOLOv4 algorithm with the processing time of all other arena components including camera acquisition. The average duration of YOLOv4 processing was measured directly (mean = 7.92 ms, SD = 0.36 ms, [Fig 2B]) and the delays associated with the rest of the arena components were measured by turning on an LED in the arena and measuring the time until the system turned it automatically off while the entire

arena was running. Specifically, we placed an LED on the arena floor and turned it on. We measured the intensity of a single pixel in the LED light spot in each top-view camera frame and sent a command to the arena controller to turn the LED off once it was on (its intensity reached a certain threshold). We measured the latency by counting the number of frames (at 60 Hz frame rate) in which the LED was turned on. Thus, the latency measurement is quantized to 17 ms bins and represents an upper bound of the real latency (Fig 2C).

**Animal head position tracking.** To track the lizards head and determine if it is inside the reinforced area, we trained YOLOv4 [51]—a light-weight convolutional neural network. The model was used to detect the animal's head bounding box in each top-view camera frame (Fig 2A). Training was performed using a custom dataset consisting of approximately 2,000 grayscale images of *P. vitticeps* lizards from various angles, cameras, and different backgrounds, as well as approximately 1,400 grayscale images that did not contain a lizard, which were added to reduce the number of false-positive detections. Approximately 800 of these negative examples were images of birds and insects from the Imagenet dataset [87], and the rest were images taken from several empty arenas. The resulting model was able to detect the lizard head in a wide range of images with various individual lizards, camera angles, and arenas (Fig 2D). Each top-view camera frame was resized to 416 × 416 pixels and then processed with the trained model. The output of the model provides a confidence score for each bounding box. Non-max suppression with a 0.6 threshold was used to remove overlapping detected bounding boxes with lower scores [51]. Additionally, detections with scores lower than 0.8 were discarded. When the output of the non-max suppression algorithm resulted in multiple bounding boxes, we used the box closest to the previous detected one and ignored the others. We used the bounding box's center point to estimate the animal's head position in the arena.

**Validation of fine-tuned YOLOv4 model.** We evaluated the model's performance by comparing manually annotated head bounding boxes and model-derived bounding boxes. We used a validation set of 400 images sampled uniformly from 800 h of experimental videos from 4 lizards. Using a confidence threshold of 80%, the model achieved a recall of 100% (no false positives) and a precision (true positive/total positive) of 78% (Fig 2D). To estimate bounding box accuracy, we divided the intersection area of the 2 boxes by the area of their union (Intersection over Union, IoU). Out of the true positive detections, the average IoU between the predicted and manually annotated bounding boxes was 73% (S1 Fig) indicating a good accuracy.

**Travel distance measurements.** We measured the travel distance of the animal by cumulatively summing the distances between head position coordinates along pairs of consecutive video frames. To bridge frames with poor YOLOv4 annotation (see above), we linearly interpolated the position in these frames using values in neighboring frames. Lizards can spend long periods of time in the same location. During these times, small random fluctuations in YOLOv4 position estimation are accumulated resulting in increased travel distance. To remove such contributions, we used Gaussian filtering on the position coordinates (window size = 51; SD = 17). The filter window size was chosen empirically by plotting travel distance as a function of window size for a range of values during periods of limited movement: Travel distance was observed to drop quickly for small window sizes and nearly saturate for windows larger than 51. To transform the measurements from pixels to centimeters, we multiplied the values by a constant factor, which was calculated by computing the distance between several pixel pairs with known physical distances and averaging the resulting multiplication factors.

**Segmentation into periods of movement and quiescence.** We segmented each experiment day into periods in which the animal moved in the arena or stayed in place based on the position-tracking data generated by the YOLOv4 model. First, we used time-based linear interpolation to replace missing position values. We applied a Gaussian filter (window size = 51, see travel distance section above) to reduce the jitter of the model output, resulting in a matrix of

$T$x2, where $T$ is the number of video frames recorded daily. Then, we created a time-delay embedding from this data (windows size = 50, gap size = 2) [88] and calculated the differences between consecutive columns for each row, resulting in a matrix of $T$x100 (each row containing 50 interleaved 2D velocity vectors). We conducted a principal component analysis on this matrix and kept the first 6 principal components (explaining more than 90% of the data variance in an animal), which resulted in a $T$x6 matrix. We then converted this matrix into a binary vector of size $T$ by calculating the L2-norm of each row and applying a threshold. A value of 1 indicated a movement sample and 0 indicated a stationary sample. This vector was then used as an input to Kleinberg's burst detection algorithm [89] to segment into periods of stationarity and movement.

**Entry rate statistics.** To measure whether animals significantly increased their entry rate to the reinforced area compared to the previous block (ΔER), we defined 96 areas (including both second and third reinforced areas) in an evenly spaced grid across the arena floor, each with the same size and shape as the reinforced areas (S5 Fig). We used the animal's position data to estimate the entry rate for each area in each block, resulting in ΔER values for each area and block transition. To calculate the entry rate for each area, we used offline simulations, assuming each time that a different area was the reinforced one and incorporating all rules for releasing a reward used in the actual task (as specified in the Results section). For correlation analysis, areas were classified as either neighboring the feeder area, the first reinforced area, the second reinforced area, or not neighboring any reinforced areas (green, orange, blue, and gray areas in S5 Fig, respectively). This was determined based on whether areas had overlapping sections with the reinforced areas; however, the classification was slightly altered to ensure areas were only considered neighbors of a single reinforced area and maintaining an equal number of neighbors for the second and the third reinforced areas. The simulation output was the number of rewards per block. We divided this value by the number of light hours in each block to derive reward rates.

**Basking periods estimation.** Animals' basking periods were determined based on position data (from YOLOv4). Animals were considered basking when their head bounding-box centroid resided within a radius of 20 cm around the heat lamp center.

**Camera thermal measurements.** We used a thermal camera (see hardware) to estimate ambient and animal skin temperatures during experiment sessions. We recovered temperature data for each pixel by decoding video files and linearly scaling the decoded 8bit images to Celsius values. We then calculated the ambient arena temperature for each frame by averaging over all image pixels. This measurement exhibited the same trend as the ambient temperature sensor on the arena wall but was 1.67°C higher on average (SD = 0.16) due to the inclusion of the basking area in the calculation.

**Animal thermal measurements.** We estimated animals' body temperature as follows. We found the temporally closest regular camera frame for each thermal video frame and extracted the animal head bounding box centroid using YOLOv4. We used the centroid coordinates as an input prompt to the SAM segmentation model [53] and produced a segmentation mask containing all animal pixels. We then linearly transformed coordinates to shift from the visual camera to the thermal camera arena coordinates. This transformation was based on a set of 112 manually labeled reference point pairs to align the centroid and mask with the coordinate space of the thermal image. We used SAM again to generate a second segmentation mask of the animal in the thermal image using the transformed centroid as input. The transformed mask was then passed through 2 iterations of erosion and was finally used to determine the estimated body temperatures by computing the median intensity across all mask pixels.

To analyze temperature dynamics over entire days, we extracted thermal video frames in 3-s intervals and executed the procedure described above for each frame. To make sure body

temperature measurements are accurate, masks that were unrealistically large or small were discarded. Specifically, only areas of 800 to 4,000 pixels for thermal image masks and 11,000 to 30,000 pixels for regular camera image masks were kept. Additionally, due to the low acquisition frequency of thermal images, pairs of regular and thermal images were occasionally spatially non-overlapping, especially during fast movement bouts. To solve this issue, we calculated the IoU between each pair of masks and removed frames in which the IoU was below 0.3 (2% of the frames; M = 0.5; SD = 0.11). Finally, we filtered the sequence of estimated skin temperatures extracted from valid frames using a moving average (window size = 51). To illustrate the conversion from skin temperature (ST) to core temperature (CT), we used the calibration in [64]. Specifically, core temperature was estimated (S2C Fig) using a second order polynomial fit ($CT(T) = p1^* ST^2 + p2^* ST + p3$, p1 = 0.0159; p2 = 1.7066, p3 = −6.1034).

**Validation of thermal body masks.**   We validated the generated thermal body masks by examining the temperature gradients across mask edge points. Since lizards were warmer than ambient temperature, these gradients are expected to sharply decrease at lizard edges. We used a uniform sample of body masks comprising approximately 10% of the total masks generated during the measurement day. Near the heat lamp, the ambient temperatures are high and it is harder to evaluate gradients. We therefore removed frames in which lizards partially overlapped with the basking area (24% of the original sample). We proceeded by calculating the temperature gradient along line segments originating from the animal mask center of mass, and extending outwards such that each edge point was at the center of its respective segment (S2A Fig). Line segments were discarded when their inner part (from animal center to edge) was not fully contained in the mask or when their outer part (from the edge until the line endpoint) overlapped with mask points. For each segment, we calculated the intensity of each pixel according to the thermal image. We aligned the segments according to their middle points and averaged to produce a mean gradient for each frame. Finally, we calculated the distribution of intensities as a function of distance along the line (S2B Fig). We aligned the average mask segments in the same way, normalized their intensities to z-score units, and calculated the median gradient across the masks (S2B Fig, red line).

## ReptiLearn hardware and arena

**Arena cage.**   The arena was shaped like a box without its top face. It consisted of a frame built with square aluminum profiles and walls and floor made from 3 mm thick aluminum composite panels. The floor dimensions were 70 cm by 100 cm, and its height was 45 cm. An additional 100 cm long profile was placed 1 m above the centerline of the arena floor using additional profiles to support the top-view and thermal cameras, as well as the grid of heat lamps (described below). S1 Table contains the price list of arena components.

**Arena computer.**   The arena hardware was connected to a desktop computer placed next to the arena, on which the ReptiLearn software was running (Intel Core i7-11700K CPU, 32GB DDR4 memory, NVIDIA GEFORCE RTX 3080 Ti GPU, 500GB SAMSUNG 980 M.2 NVME SSD, and a 2TB 7200 RPM HDD). An Ubuntu 22.04 Linux operating system was used.

**Video cameras.**   Five cameras were used to acquire video and thermal data in the arena. They were attached to the arena using short adjustable arms (Noga Engineering & Technology, LC6100). Three FLIR Firefly S USB3 monochrome cameras (FFY-U3-16S2M-S) with 6 mm lens (Boowon BW60BLF) were attached to profiles at the top edges of the arena walls. A top-down view of the arena floor was captured using a FLIR Blackfly S USB3 color camera (BFS-U3-16S2C-CS) with a 2.8 to 10 mm varifocal lens (Computar A4Z2812CS-MPIR). Although this is a color camera, only monochrome data was acquired. The camera was attached to the middle of the center-top profile and was used for real-time position tracking.

To prevent IR light emitted by heat lamps from flooding images, we covered the Firefly camera lens with an IR cutoff filter attached using custom 3D-printed holders. Video was mainly recorded at a frame rate of 50 Hz.

**Thermal camera.**   We used a thermal IR camera (FLIR A70) positioned next to the Blackfly camera above the arena to measure temperature dynamics during daily activity periods. The camera captures images in a resolution of 640 × 480 pixels at 16 bits per pixel (bpp), representing temperatures in up to 10 mK resolution. The camera thermal resolution is advertised as 45 mK or less; however, we linearly scaled each image to 8 bpp and encoded it into video files to store the thermal data. We chose a temperature range between 20˚C and 45˚C and scaled accordingly, which resulted in approximately 0.1˚C resolution. The camera supports up to 30 frames per second; however, images were taken at a frequency of about 3 Hz due to technical issues, which FLIR developers eventually solved (but only after we finished conducting the experiments).

**Touch screen.**   A touch screen (ELO Touch Solutions AccuTouch 1790L 17” LCD open frame) was attached to one of the shorter arena walls using screws. The touch screen was connected to the arena computer using HDMI and USB cables. A cardboard was fitted around the screen to prevent animals from using it to leave the arena.

**Arduino microcontroller boards.**   Three Arduino boards (Arduino Nano Every) were used to control the arena lighting, food dispensers, temperature sensors, and heat lamps and send TTL signals to synchronize the video cameras (Figs 1B and S6). The boards were connected to the arena computer using USB cables and placed inside a box attached to the external side of an arena wall.

**Light.**   The arena was lit using an LED strip (12V, 6500K white LEDs, approximately 3.4 m long) that was attached using adhesive to profiles at the top edge of the 4 arena walls. This provided relatively uniform lighting across the arena, minimizing shadows. The strip was controlled using a relay module (based on an Omron G5LE-14-DC5 5VDC SPDT relay). The module's EN, VDD, and GND control ports were connected to one of the Arduino boards, and it was used to control the DC output of the LED strip 12V, 5A power supply unit (S6A Fig).

**Live prey dispenser.**   We used a widely available aquarium food dispenser (EVNICE EV200GW) for rewarding animals with live prey. It was attached to the arena frame using the included clamp and released rewards into a small dish placed on the arena floor below it. Commercially available aquarium dispensers are suitable for providing live food. However, they are slow and do not provide means for external control. To control the food dispenser and reduce its latency, we modified its control circuit. We connected a ULN2003 stepper motor driver board to the feeder's 28BYJ-48 stepper motor (see S6A Fig for circuit details). An Arduino board controls the motor driver, and Arduino code, integrated into the arena controller, implements an alternative motor control sequence, significantly reducing its latency to 3.21 s (SD = 0.54 s) (Fig 2F). This was measured by calculating the time difference from sending a command to the worm dispenser until the dispensed worm was detected hitting the arena floor in the video stream. Feeders were filled with worms each kept in a small compartment (15 in total) with food for gut loading the worms.

To keep animals motivated, smaller worms are more suitable. We therefore used *T. molitor* larvae as rewards. However, these worms may turn to pupa in experiments with low reward rate and long durations of a few days. In such cases, using young *Zophobas atratus* can extend larval stages [90]. In some experiments, a larger quantity of worms was needed between refills. This was solved by vertically and horizontally stacking the feeders (up to 30 worms in 2 vertically stacked feeders and 150 for 5 horizontally stacked feeders). We vertically aligned stacked feeders such that the top feeder released its reward through the release hole of the feeder below

it. The experiment module tracked the number of available rewards in each feeder and determined which device should discharge a reward accordingly. After resupplying the feeders, researchers can notify the software remotely using the Web UI. Multiple feeders can be placed in the arena and they require no manual intervention except loading every few days.

**Heat grid.** Twelve infrared halogen heat lamp bulbs (24 volts, 50 watts) were attached above the arena, arranged in an equally spaced grid of 3 rows by 4 columns. We attached 8 flat slotted steel bars to the center-top profile to hold the lamps. Lamp heating can vary between different manufacturers resulting in different ground temperatures. To control heating intensity, we modified lamp height by attaching 20 cm steel M6 spacer to the steel bars. The lamp was attached to a ceramic G6.35 socket using appropriate holders and screws. Lowering the lamps towards the arena floor, increased temperature and reduced the heating area of each lamp. The lamps were connected to a dedicated Arduino board through a 16-channel relay module board (S6B Fig). The relay board routed the output of a 12A, 24V DC power supply unit to specific lamps, which we used to simultaneously operate up to 2 lamps. We selected this high maximum current capacity to prevent potential problems caused by the high inrush current of halogen filaments. Another relay module (based on an Omron G5LE-14-DC5 5VDC SPDT relay) controlled the power supply unit output as an additional safety measure (S6A Fig). The grid enabled targeted heating of specific arena regions by more than 10˚C, covering nearly the entire arena floor (Fig 3A).

**Temperature sensors.** We used 2 plastic-covered digital DS18B20 temperature sensors to measure temperature conditions in the arena. One sensor was attached to the back wall of the arena and measured ambient temperatures. The second sensor was placed on the bottom of the opposite wall under the basking area heat lamp. The sensors were connected to one of the Arduino boards using a 1-Wire connection (S6A Fig).

## ReptiLearn software

**General design.** The ReptiLearn software, written in Python, provides a toolkit for automating closed-loop behavioral tasks, collecting behavioral data, and extracting basic behavioral features. It includes a customizable real-time image processing pipeline that can be used to process and record synchronized video data from multiple cameras or other image sources. An arena controller program provides a generic way to integrate custom hardware components into the system without writing code by communicating with any number of Arduino microcontroller boards (S7 Fig). Controlling and monitoring the system can be done remotely through a web-based user interface (Web UI; Fig 1D). Users can implement new automated experiments by writing Python scripts and linking them to experiment sessions. These scripts can automate any part of the system based on real-time information gathered from arena sources. Non-programmers can customize existing scripts by modifying session parameters through the Web UI. A scheme of blocks and trials makes it possible to design complex experimental sequences.

**Parallelism and synchronization.** To overcome limitations in Python's concurrency model, the software makes extensive use of separate OS processes (S7 Fig). These processes are synchronized using a central state store, which holds the current state of all system components in one place. The Web UI and other external applications can receive updates whenever the state data changes by making a WebSocket connection to the system HTTP server. The server also provides an API to control the system remotely, and the MQTT protocol is supported for communicating with external devices and software (described below).

**Supported operating systems and license.** The software can run on a wide range of operating systems thanks to Python's cross-platform support. It was tested on Ubuntu 20.04,

Ubuntu 22.04, and recent versions of Microsoft Windows and macOS. The software is licensed under the open-source GPL-3.0 license. Source code, detailed installation instructions, and guides for running and adapting the system are available at https://github.com/EvolutionaryNeuralCodingLab/reptiLearn.

**Web-based user interface.** The Web UI is implemented as a separate JavaScript application using the React framework (code is available at /ui in the GitHub repository). After an initial build process (described in the "Getting Started" guide), the system HTTP server can be used to access it from any device that includes a modern web browser (Figs 1D and S8). It provides live video streaming from multiple sources and communicates with all other system components through a WebSocket connection and an HTTP API. The various features of the interface are described below in relevant sections.

**Video acquisition software and image sources.** Image data from multiple sources can be acquired using the system for real-time processing and offline analysis (S9 Fig). ImageSource classes collect raw data from cameras or other imaging devices and make it available for further processing. Support for FLIR cameras and potentially other GenICam cameras are provided by the FLIRImageSource class using the Spinnaker SDK Python bindings. Allied Vision cameras are also supported through the AlliedVisionImageSource class, which utilizes the Vimba SDK. Additionally, video files can be used as sources for simulation and debugging purposes using the VideoImageSource class. This class also supports capturing images from standard webcams. Additional sources can be supported by writing new ImageSource classes and storing them in Python modules (inside the /system/image_sources folder).

**Image observer.** ImageObserver classes provide an interface for further processing image data acquired by ImageSource classes. Each ImageObserver is attached to an ImageSource and is notified whenever a new image is acquired. Similarly to image sources, observer classes found inside the /system/image_observers folder are automatically recognized. The repository includes a YOLOv4ImageObserver class that performs object detection and can generate bounding boxes for objects or animals in the arena (the YOLO model weights are not included; see "Getting Started" guide). Additional simple observer classes are included as examples for implementing new processing algorithms. Multiple ImageObserver classes can be attached to the same ImageSource, and an additional VideoWriter observer is attached by default to each source for recording and encoding video. Both sources and observers can be added and configured using the Web UI "Video Settings" window (S8A Fig). When adding new sources and observers, all available classes found in their respective folders are listed, and users can modify classes and reload them without needing to restart the software. ImageObserver output can be accessed from the experiment script using a provided API.

**Parallel image processing.** To improve performance and utilize multiple CPU cores or GPUs, each ImageSource and ImageObserver instance runs in a separate OS process and communicates with other components through shared memory buffers to avoid expensive data copying (S9 Fig). There is no limit to the number of sources and observers that can be used concurrently. Using this architecture, data from each image source can be simultaneously used for encoding video, streaming video over HTTP, and real-time processing by multiple algorithms with minimal latency.

**Video encoding.** Video is encoded by the FFmpeg library using the ImageIO python library. Encoding profiles can be defined in the system configuration file and selected in the "Video Settings" window. For encoding using NVIDIA GPUs, we used the NVENC encoder; however, any encoder can be configured by setting FFmpeg parameters accordingly. Specific FFmpeg builds can be used by pointing the ImageIO library to a particular FFmpeg executable (see "Getting Started" guide). In addition to video files, each recording includes a CSV file containing a timestamp for each frame and a JSON file containing metadata about the recording.

Timing data is supplied by the ImageSource for each frame and, in supporting sources, represents the camera exposure start time.

**Camera synchronization.** Camera synchronization is accomplished by connecting the output pin of an Arduino board to the GPIO input of supporting cameras (S6C Fig). A trigger interface is provided by the arena controller for this purpose (see Arena controller) and does not require programming the Arduino manually. The video system automatically identifies an existing trigger and provides manual control through the Web UI. Additionally, the trigger is automatically paused for 1 s before starting a recording to simplify synchronization with external sources, such as electrophysiological data.

**Arena controller.** Communicating with the various arena electronic components (described above) is done using the arena controller program (found in the /arena folder of the repository). The program is integrated into the ReptiLearn system; however, it can also be used standalone or even on a different computer. It maintains two-way communication between the rest of the system and any number of Arduino boards by relaying commands received over an MQTT connection to serial protocol over USB, as well as forwarding data received from Arduino boards over designated MQTT topics using a simple JSON-based protocol. A single Arduino program includes all the code necessary to operate a wide range of devices, avoiding the need to program the boards manually. The "Arena Settings" window in the Web UI provides an interface to configure the controller, identify connected boards, and upload the unified Arduino program to each board.

Each device connected to an Arduino board is controlled by an interface class. Several interface classes are implemented, and more can be added by implementing them using C++. For example, the LineInterface is used to control a single digital output (e.g., for switching light sources), the FeederInterface controls the automatic feeder (described above), and the TriggerInterface is used for sending TTL pulses at a selected frequency to synchronize camera acquisition. Each interface provides specific configuration parameters, has a current state value (e.g., a measurement or whether it is turned on or off), and can respond to multiple commands (see docs/arena_interfaces.md for more details).

Once configured, the ReptiLearn system integrates with the controller in several ways. The arena module (at system/arena.py) is responsible for executing the controller program on startup and provides functions for communicating with it, which can be used to control interfaces from experiment scripts. It also maintains a list of all current interface values in the state store, which is updated after each interface command is sent, and by polling the interfaces at a fixed interval (once a minute, by default). Additionally, it can be configured to store interface values (such as temperature sensor measurements) in CSV files or a database using a data logger (see data collection below). The Web UI provides an Arena menu (S8B Fig) for manual interaction with the controller, where individual interfaces are listed with their current state. It can also be used to control the execution of the controller program and to send commands to interfaces, for example, to trigger a reward feeder or toggle a digital output manually. Once the controller is configured, the same configuration file is automatically used to generate this menu.

**Touchscreen interface.** A web application is included to display stimuli on screens and receive touch input into the system (at /canvas). The application uses the Konva.js 2d canvas JavaScript library and exposes large parts of its API over a bidirectional MQTT connection. The canvas module (/system/canvas.py) provides all the necessary functionality to communicate with the web application through the Canvas class. This architecture supports multiple screens by communicating with multiple web app instances, possibly running on different computers. Canvas classes can be used in experiment scripts to display various objects, such as shapes, images, or videos, and manipulate and animate properties of these objects (e.g., their

position or color). Scripts can be notified of various events, such as screen touches of specific objects, video and animation progress, and other events supported by Konva.js. Example experiment scripts that use this module are available in the repository (see /system/experiments/canvas_shapes.py, for example).

**Data collection.** The software provides several methods for collecting and storing data for offline analysis. When starting a new session, a session folder is created inside the session root folder (as defined in the configuration file), in which all video, image, and data files are stored. Data loggers, implemented by the DataLogger class (see /system/data_log.py), can be used to store data in CSV files as well as in TimescaleDB database tables, which can also be used to provide real-time visualization of data using third-party applications. Each data logger runs in a separate OS process to ensure that data collection does not interfere with other parts of the system.

Several specialized data loggers are provided: (1) An event logger (see /system/event_log.py) automatically keeps track of session events and can be further configured to log changes of specific state store values or incoming MQTT messages with specific topics. Additionally, experiment scripts can use the event logger to log any event relevant to the experimental paradigm. (2) ImageObserver loggers (ObserverLogger class) can be configured to collect the output of a specific observer (for example, to record animal position for each video frame). (3) One can define custom loggers within scripts using a generic API to record data from any source, such as screen touches or the position of objects on the screen (for example, see /system/experiments/canvas_video.py). An offline analysis module (at /system/analysis.py) provides classes and functions for analyzing the data stored in session folders, simplifying tasks such as finding video frames matching a specific session event or reading time-series data created by data loggers.

**Failure recovery.** Ensuring the ability to quickly recover in the face of unavoidable interruptions, such as power or system failures, is crucial for the success of long-term experiments. Consequently, data loggers were designed to instantly save all gathered data to the session folder. The session state is periodically recorded in JSON format, triggered by significant events like the initiation and conclusion of trials and blocks. Furthermore, sessions can be paused and resumed at a later time without introducing any disruptions or complexities to the analysis process.

**Creating and running automated experiments.** As described above, each system feature can be automated by implementing and using experiment scripts. The experiment module (at /system/experiment.py) includes an Experiment class that provides hooks for triggering code at various events: when the session is loaded (setup method), when a trial or block begins or ends (run_trial, end_trial, run_block, and end_block methods), and when the session is closed (release method). Sessions can be created and configured to run a specific script using the "New Session" dialog in the Web UI (S8C Fig). Any Experiment class found inside the experiments folder (at /system/experiments) is automatically listed and can be used in a session. Example scripts and scripts that were used in this study can be found in this directory of the GitHub repository.

Scripts can define default parameters, which can then be modified using the session section of the UI once a session is opened (S8D Fig). The session section also allows defining experiment blocks, each having different parameter values. Additionally, several built-in parameters are provided for controlling the number and duration of trials in each block, as well as block and inter-trial interval durations ($num_trials, $trial_duration, $block_duration, and $inter_trial_interval, respectively). Experiment classes can also define custom actions that can be triggered using the Web UI.

The system simplifies the process of developing and testing experiment scripts. Log messages generated by the code become instantly accessible within the Web UI, and updating the

code following modifications does not necessitate a system restart. This results in a swift and efficient code-test-debug feedback cycle. Since ReptiLearn is developed purely in Python, scripts can access all of the system code and interface with additional Python packages as needed.

**Task scheduling.** Controlling the timing of code execution is an integral part of running automated experiments. Consequently, the system provides several ways to schedule tasks. Scripts can utilize the schedule module (at /system/schedule.py) to set timers and trigger functions to run at specific times each day or at regular intervals. The asyncio python library is also supported and can be used for similar purposes. Additionally, task functions can be defined inside the tasks folder (at /system/tasks) to be scheduled manually using the Schedule menu in the Web UI.

## Supporting information

**S1 Fig. YOLO4 bounding box accuracy.** YOLOv4 intersection-over-union (IoU) distribution over a validation set consisting of 400 images sampled uniformly from video data of 4 animals and indicating good overlap with animal head. Individual numerical values are provided in S1 Data.
(PDF)

**S2 Fig. Validation of animal thermal body mask and core temperature estimation. (A)** Construction of line segments. An image of a lizard taken from the thermal camera is shown at the background. Line segments (white) were extended from the animal mask's center of mass (orange) outwards to the direction of each animal edge point (green). The length of each line segment was twice the distance from the center of mass and each edge point (white dots). **(B)** Density plot of the temperature at each distance along the line segments. The red line shows the median temperature gradient across all frames. A sample of uniformly selected 346 thermal video frames measured for 1 day is analyzed. **(C)** Movement dynamics (travel speed, blue) and corresponding estimated core temperatures (red, with ambient temperature in orange) as well as reward times (green) and basking periods (gray) measured over a single day (taken from Fig 3D). Individual numerical values are provided in S1 Data.
(PDF)

**S3 Fig. Reward rate as a function of daytime for different animals.** Reward times were collected (as in Fig 4C), convolved with a normalized Gaussian (std = 30 min) and averaged over all experiment days. Individual numerical values are provided in S1 Data.
(PDF)

**S4 Fig. Progression of spatial correlations in entry rate over days. (A)** Correlation coefficient of ΔER decay as a function of distance from reinforced area 1 (as in Fig 4G), calculated for each day separately (day zero marks the first day of area 2 reinforcement, gray shade marks days before reinforcement of area 1). Black line marks the average over animals. **(B)** Same as (A) but for reinforced area 2. Animal 2 did not complete the reversal to area 2 and was excluded from this analysis. Individual numerical values are provided in S1 Data.
(PDF)

**S5 Fig. Locations of 96 areas used for entry rate analysis.** Each dot represents the center of an area with the same shape and size as the reinforced areas. Green, orange, and blue circles show the area of the feeder, the second and the third reinforced areas, respectively. Colored rectangles represent the areas neighboring each of the reinforced areas (marked in Fig 4G–4I).
(PDF)

**S6 Fig. Circuit diagrams of arena electronic components. (A)** An Arduino board connected to 2 feeders, 2 temperature sensors, an LED strip relay module, a cue LED, and a relay module controlling the heat grid's power supply unit. **(B)** A second Arduino board connected to a 16-channel relay module that controls each of the 12 heat lamps individually. **(C)** A third Arduino board was responsible for synchronizing image acquisition by sending TTL pulses to GPIO inputs of 4 cameras in parallel.
(PDF)

**S7 Fig. ReptiLearn software architecture.** The software consists of an image processing and video recording system, an Experiment class controlling the current experiment session, a state store used for synchronizing different processes, data loggers, and an MQTT client responsible for communicating with the arena controller, touch screen app, and other external software. The HTTP/WebSocket server facilitates real-time monitoring and control of the software. The arena controller handles communication with Arduino boards that control arena hardware components.
(PDF)

**S8 Fig. ReptiLearn Web UI. (A)** Video settings window showing the parameters of an Image-Source. **(B)** New session dialog. The session uses the spatial learning Experiment class found in module/system/experiments/loclearn2.py. Session id determines the directory name in which data is to be stored. **(C)** The arena menu listing every configured arena controller interface. Feeder items can be clicked on to release a reward. Toggle interface items can be switched on or off. Sensor interface items display their most current measurement. **(D)** Session UI section displaying the current session name and the time of creation at the top. Located below is the session control bar that allows to start and stop the experiment and to control the current trial and block. Session and block parameters can be set using the editors in the bottom tabs.
(PDF)

**S9 Fig. Video system data pipeline diagram.** ImageSource objects acquire images and store them in shared memory buffers (green) together with timestamps. Each ImageObserver object is tied to an ImageSource and is notified when new data is written to the shared buffer (green). It processes the data and outputs a result to another shared buffer (blue). The Experiment class can then access these data through a simple API. ObserverLogger objects can access ImageObserver buffers directly and log any new results to a file or database. VideoWriter objects are specialized ImageObservers that encode and write ImageSource buffer data to video files.
(PDF)

**S1 Data. Excel sheet with individual data points presented in the figures.** The excel sheet contains different tabs, each including data for individual data points in all relevant panels of a specific figure.
(XLSX)

**S1 Table. Statistics for entry rate correlations across animals.** Mann–Whitney U statistics for the distributions of correlation coefficients of the difference in entry rate as a function of distance from each area, for all simulated and real areas. Feeder areas were excluded from the correlation calculation.
(DOCX)

**S2 Table. Price list for arena components.** All components, including the visible light cameras, are relatively low cost. An exception to this is the PC which is necessary to facilitate real-time processing. Another exception is the thermal camera that is not required if thermal

monitoring is not a part of the experimental design.
(DOCX)

## Acknowledgments

The authors are most grateful to R. Eyal for guidance during the initial phase of the project; A. Shvartsman for technical and administrative assistance; the animal caretaker crew for lizard care; the Shein-Idelson laboratory for their suggestions during this work; and F. Baier for comments on the manuscript.

## Author Contributions

**Conceptualization:** Mark Shein-Idelson.

**Formal analysis:** Tal Eisenberg.

**Funding acquisition:** Mark Shein-Idelson.

**Investigation:** Tal Eisenberg.

**Methodology:** Tal Eisenberg.

**Software:** Tal Eisenberg.

**Supervision:** Mark Shein-Idelson.

**Visualization:** Mark Shein-Idelson.

**Writing – original draft:** Tal Eisenberg.

**Writing – review & editing:** Mark Shein-Idelson.

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
