## [Editor Report · Decision Letter 0]

31 Oct 2023

Dear Dr Shein-Idelson, 

Thank you for submitting your manuscript entitled "ReptiLearn: A Smart Home Cage for Behavioral Experiments in Reptiles" for consideration as a Methods and Resources Article by PLOS Biology. Please accept my sincere apologies for the delay in getting back to you as we consulted with an academic editor about your submission.

Your manuscript has now been evaluated by the PLOS Biology editorial staff, as well as by an academic editor with relevant expertise, and I am writing to let you know that we would like to send your submission out for external peer review.

Once your full submission is complete, your paper will undergo a series of checks in preparation for peer review. After your manuscript has passed the checks it will be sent out for review. To provide the metadata for your submission, please Login to Editorial Manager (https://www.editorialmanager.com/pbiology) within two working days, i.e. by Nov 02 2023 11:59PM.

Kind regards,

Richard

Richard Hodge, PhD

rhodge@plos.org

PLOS

---

## [Decision Letter · Decision Letter 1]

15 Dec 2023

Dear Dr Shein-Idelson,

Thank you for your patience while your manuscript "ReptiLearn: A Smart Home Cage for Behavioral Experiments in Reptiles" went through peer-review at PLOS Biology as a Methods and Resources Article. Please accept my sincere apologies for the long delays that you have experienced during the peer review process. Your manuscript has now been evaluated by the PLOS Biology editors, an Academic Editor with relevant expertise, and by three independent reviewers.

In light of the reviews, which you will find at the end of this email, we are pleased to offer you the opportunity to address the comments from the reviewers in a revision that we anticipate should not take you very long. This includes restructuring the organization of the manuscript, reframing the Introduction section and providing additional reporting details for the statistical analyses. After discussions with the Academic Editor, we also ask that the revision provides a platform to ease the replication of the set-up to increase its accessibility/user-friendliness, since the system may be difficult for a non-engineer to implement. 

We will then assess your revised manuscript and your response to the reviewers' comments with our Academic Editor aiming to avoid further rounds of peer-review, although might need to consult with the reviewers, depending on the nature of the revisions.

*IMPORTANT*

In addition, we would be grateful if you could please provide address the following editorial and data-related requests that I have provided below (A-E):

(A) We would like to suggest the following modification to the title: 

“ReptiLearn: an automated home cage system for behavioral experiments in reptiles without human intervention”

(B) You may be aware of the PLOS Data Policy, which requires that all data be made available without restriction: http://journals.plos.org/plosbiology/s/data-availability. For more information, please also see this editorial: http://dx.doi.org/10.1371/journal.pbio.1001797

-Supplementary files (e.g., excel). Please ensure that all data files are uploaded as 'Supporting Information' and are invariably referred to (in the manuscript, figure legends, and the Description field when uploading your files) using the following format verbatim: S1 Data, S2 Data, etc. Multiple panels of a single or even several figures can be included as multiple sheets in one excel file that is saved using exactly the following convention: S1_Data.xlsx (using an underscore).

-Deposition in a publicly available repository. Please also provide the accession code or a reviewer link so that we may view your data before publication. 

Figure 2B-C, 2E-F, 3B, 3D, 4G-I, S1B, S2

(C) Thank you for already depositing the source code in Github (https://github.com/EvolutionaryNeuralCodingLab/reptiLearn). We ask that you please link the code deposition to the Zenodo repository (https://zenodo.org/) to ensure that the deposition is given a DOI and long-term maintenance. 

(D) Please also ensure that each of the relevant figure legends in your manuscript include information on *WHERE THE UNDERLYING DATA CAN BE FOUND*, and ensure your supplemental data file/s has a legend.

(E) Please ensure that your Data Statement in the submission system accurately describes where your data can be found and is in final format, as it will be published as written there. 

We expect to receive your revised manuscript within 2 months. Please email us (plosbiology@plos.org) if you have any questions or concerns, or would like to request an extension. 

**IMPORTANT - SUBMITTING YOUR REVISION**

*Resubmission Checklist*

*Published Peer Review*

*PLOS Data Policy*

*Blot and Gel Data Policy*

Sincerely,

Richard

Richard Hodge, PhD

rhodge@plos.org

REVIEWS:

Reviewer #1: This is a methods paper that introduces and details a new automated system: ReptiLearn, for measuring behaviour, cognition and thermal physiology of ectotherms in the lab. Overall the paper is very well written, with few errors. It's also well detailed. The authors introduce a very impressive system that is a first of its kind for ectotherms. In reptile cognition, we are constrained by an animal model system in which animals can process relatively few food rewards in a day. Their behaviour is also constrained by temperature and lots of human intervention to set up and run individual trials. The system they describe can certainly change things and they demonstrate this with 4 bearded dragons that are used in cognitive trials and for measuring behaviour. I think this is a valuable system with lots of promise and is novel for ectotherms.

The only major comment I have is to consider the structure of the methods and results. It's tricky given the nature of this study, but I found the results to be a real mixture of methods and results. I suggest adding subheadings to make the paper easier to follow and restructuring methods and results in a more traditional format.

I have the following relatively minor suggestions.

Abstract, P9

"In contrast, natural behaviors evolve over multiple time scales and under minimally constrained conditions in which actions are

selected through bi-directional interactions with the environment and without human intervention." I see what you are saying but the natural world comes with a whole other set of constraints (threat of predation, varying food availability, etc). I would not use "minimally constrained conditions".

When you say "more natural experimental designs" do you mean more representative of what happens in nature?

Introduction

P 10 survival instead of survivability

I'm not sure I would say that behavioural ecology is aimed at capturing the full range of behaviours in the wild. It's more about studying behaviour in an evolutionary/fitness framework. Ethology is more likely to capture the full range of behaviours in the wild.

You mean bi-directional interactions quite frequently. It might be worth clearly explaining what you mean by this and offering some examples at the start.

1980s not 1980's

"…models such as species from the reptilian and amphibian classes." I would just say …models such as amphibians and reptiles.

"Correspondingly, the cognitive capacities of reptiles and amphibians remain poorly understood and research linking behavior with neurophysiology is scarce (39).." You could also cite Szabo et al review of reptile cognition in Biological Reviews.

In the intro you could reference some of the automated cognition work going on in the wild, such as Lucy Aplin's work on tits.

Results

Pg 1. The first sentence has no relationship to the sentences that follow. Need some restructuring.

How about subheadings in the Results? It would make the paper easier to follow and give it a more defined structure.

Larvae not larva

Pg 12 Results. Postures can also be related to temperature, they can be influenced by environment.

P 14, pg 1 have a look at Devi Stuart-Fox's publications on Pgonoa temperature using thermal cameras. I'm also not clearly understanding how you verified body temperature? A common method is to measure cloacal temperature and skin temperature at the same time and use regression to predict body temperature from skin temperature.

"ReptiLearn also includes hardware and software for interacting with reptiles visually and via touch (58)." Be specific by what you mean when referring to interaction via touch. 

P 15 you say 2 s and then 5 seconds—be consistent.

The results section has a lot of methods mixed in. For example, p 15 pg 2. You may need to carefully consider the methods and results and rework and restructure them to be more complementary.

Instead of saying Pogona you should say P. vitticeps.

P 16, nonsignificant, not "insignificant".

Fig. 1 d. Hopefully this will be bigger in publication or that we can open into a bigger format. As presented, it's too small to see much or read any of the text.

Fig. 4. Is the yellow area actually demarcated in the arena or is it spatially mapped on the camera system?

Discussion

Section on temperature, p 21-22. In reptile studies, it's also very important to know an animal's preferred body temperature because this is crucial for designing behavioural and cognitive studies and for understanding its natural history. We typically do this in a thermal gradient from very hold to very hot and measure the animals temperature using a probe/data logger, etc. You could tout your system more directly as an alternative method of measuring Tpref. You kind of already say this but not explicitly. (There is a quite vast literature on lizard thermal biology.) In thermal gradients they don't surgically implant sensors but they do insert probes into the cloaca.

Are there other cognitive tests that you think ReptiLearn could be used for?

Bottom of p 22 and maybe other places. It's better not to use "cold-blooded" because many ectotherms, such as lizards, have high preferred body temperatures, sometimes in excess of 40 C.

P 22, bottom "Despite pioneering work in the field.." I would cite Szabo et al review of reptile cognition in Biological Reviews.

Methods section

Spatial learning task. Can you provide details of the actual task? The majority of the details are in Figure 4 in Results and many of the details are also scattered in results. 

You say it is low cost. I understand that costing will vary based on types of cameras, etc., but it would be good to have a ballpark of what that would be.

Reviewer #2: I was very excited to see this manuscript which presents an automated approach to assess reptile behaviour. This is both a timely and innovative approach to studying reptile cognition and I enthusiastically support publication of the manuscript subject to some minor changes.

1. The introduction presents a rather old fashioned view of how different fields approaches behaviour and cognition. I believe most of these fields see the benefit from the other, often collaborate and the introduction should be set up to frame the key challenges rather than different approaches - all approaches would benefit from this methodological advancement.

2. I would like to see the data from the spatial task and reversal presented in a traditional way so researchers can see clearly how the data compares to standard data, particularly in relation to learning rates. Consideration of how long this sort of data might take to collect using a standard paradigm would also be extremely useful.

Reviewer #3: This well-written manuscript introduces a semi-automated software and cage set-up designed for reptiles and demonstrates its utility in learning tasks in Pogona lizards. 

Introduction

1. The introduction effectively sets the stage for the need for ReptiLearn. The historical context and the limitations of existing methodologies are well articulated. However, the authors could further sharpen this section by briefly mentioning earlier attempts (if any) to automate reptile behavior studies and how ReptiLearn improves upon these.

2. The Methods section is quite detailed, which is commendable. However, it might benefit from additional subheadings or bullet points for better readability, especially in outlining the technical specifications of ReptiLearn.

Technical specifications

1. The technical description of ReptiLearn, including the software, hardware, and integration aspects, is thorough. However, it would be helpful to include a discussion on potential technical challenges or limitations one might have encountered and how they were addressed.

2. In the results section, the data on Pogona vitticeps is compelling. However, the paper would benefit from additional discussion on the generalizability of these results to other reptile species, given the diversity within this class.

3. The integration of temperature control in your behavioral experiments is innovative. However, more detail on how these temperature conditions were optimized for different reptile species (if tested) would be beneficial.

4. Can the authors comment on the reward delivery latency? Is this delay appropriate for most rewarded learning tasks?

Statistical analyses

1. The analysis seems robust, but more information on the statistical methods used for data analysis, particularly in the animal head position tracking and travel distance measurements, would be useful. This would include any assumptions made in your statistical models.

2. In the Results and Discussion sections, while they have shown that ReptiLearn is effective for Pogona vitticeps, a more detailed discussion on its effectiveness for studying a broader range of behaviors in reptiles would be beneficial, especially considering the variability in behavioral patterns across different reptilian species.

Broader implications/future directions

1. The Discussion section adeptly covers the implications of findings and the potential of ReptiLearn. Expanding on how this system could pave the way for new research areas in reptile behavioral studies would enrich this section.

2. The potential for ReptiLearn's application in evolutionary biology and comparative neurology is exciting. A more detailed exploration of these possibilities, perhaps with hypothetical scenarios or proposed studies, would be valuable.

Figures:

1. Your figures are informative and well-integrated into the text.

2. For Figure 3, the gray area (corresponding to basking periods) does not appear to show up correctly in my version.

In summary, your manuscript presents a significant contribution to the field of reptile behavior studies. The development of ReptiLearn is an innovative step forward, providing a new tool for researchers in this field. With some additional details and clarifications, particularly in the Methods and Results sections, your paper could have an even greater impact.

---

## [Editor Report · Decision Letter 2]

2 Feb 2024

Dear Dr Shein-Idelson,

Thank you for the submission of your revised Methods and Resources Article "ReptiLearn: an automated home cage system for behavioral experiments in reptiles without human intervention" for publication in PLOS Biology. On behalf of my colleagues and the Academic Editor, Ann Clemens, I am pleased to say that we can in principle accept your manuscript for publication, provided you address any remaining formatting and reporting issues. These will be detailed in an email you should receive within 2-3 business days from our colleagues in the journal operations team; no action is required from you until then. Please note that we will not be able to formally accept your manuscript and schedule it for publication until you have completed any requested changes.

PRESS

Kind regards, 

Richard Hodge, PhD

rhodge@plos.org

PLOS
